# How Wearable Sensors Can Support the Research on Foetal and Pregnancy Outcomes: A Scoping Review

**DOI:** 10.3390/jpm13020218

**Published:** 2023-01-26

**Authors:** Andrea Maugeri, Martina Barchitta, Antonella Agodi

**Affiliations:** Department of Medical and Surgical Sciences and Advanced Technologies “GF Ingrassia”, University of Catania, 95123 Catania, Italy

**Keywords:** pregnancy, wearables, wearable devices, sensing, neonatal, personalized prevention

## Abstract

The application of innovative technologies, and in particular of wearable devices, can potentially transform the field of antenatal care with the aim of improving maternal and new-born health through a personalized approach. The present study undertakes a scoping review to systematically map the literature about the use wearable sensors in the research of foetal and pregnancy outcomes. Online databases were used to identify papers published between 2000–2022, from which we selected 30 studies: 9 on foetal outcomes and 21 on maternal outcomes. Included studies focused primarily on the use of wearable devices for monitoring foetal vital signs (e.g., foetal heart rate and movements) and maternal activity during pregnancy (e.g., sleep patterns and physical activity levels). There were many studies that focused on development and/or validation of wearable devices, even if often they included a limited number of pregnant women without pregnancy complications. Although their findings support the potential adoption of wearable devices for both antenatal care and research, there is still insufficient evidence to design effective interventions. Therefore, high quality research is needed to determine which and how wearable devices could support antenatal care.

## 1. Introduction

High-quality antenatal care is crucial to ensure maternal and neonatal health, along with providing evidence-based information and education on healthcare to expectant women. Although most pregnancies are uneventful, high-quality antenatal care is needed to provide timely prevention and to manage the minority that have complications [1]. A significant portion of the recent attention surrounding personalized medicine is due to the application of innovative technologies, which provide a continuous stream of data regarding physiological parameters, metabolic status, and behaviours [2,3,4]. Some of these innovative technologies—such as smart watches, bracelets, armbands, apps for mobile health, etc.—have become an integral part of our daily lives [5]. There are at least five features that make wearable devices suitable for use in healthcare: wireless mobility; interactivity; sustainability; miniaturization; and wearability [6]. If on the one hand, wearable devices can monitor vital signs, clinical parameters, and behaviours, on the other hand they also help improve health status by enabling the design of specific and personalized interventions [7]. This consideration is crucial in view of the modern approach to medicine, which should be aimed at preventing and predicting diseases and making treatments more personalized and participatory. It is also worth noting that wearable devices can play an obvious role in facilitating the emergence and advances of connected healthcare [6].

The application of innovative technologies, and in particular of wearable devices, can potentially transform the field of antenatal care with the aim of improving maternal and new-born health. This research field, however, is in continuous expansion and requires a constant updating on newest devices, research pitfalls, and potential perspectives. Although several systematic reviews exist about the role of mobile health in antenatal care [8,9,10], none of them have focused on the use of wearable devices. Yet, there is also still controversy over the efficacy of interventions based on the use of wearable devices by mothers, probably due to methodological and clinical heterogeneity among studies [6,7,9]. A scoping review was therefore conducted in order to map the research conducted using wearable sensors during pregnancy. Accordingly, the following research question was formulated: what is known from the literature about the use of wearable sensors in the research of foetal and pregnancy outcomes?

## 2. Materials and Methods

The protocol for the current scoping review was drafted using the Preferred Reporting Items for Systematic reviews and Meta-Analyses extension for Scoping Reviews (PRISMA-ScR) checklist [11]. To identify all potentially relevant articles, the following relevant medical databases were searched from 2000 to October 2022: MEDLINE, EMBASE, Web of Science, BIOSIS Citation index, Derwent Innovations index, and KCI-Korean Journal database. The search strategy consisted of the following combination of terms: ((wearable) OR (wearables) OR (mobile device)) AND ((pregnancy) OR (pregnant) OR (mother) OR (maternal) OR (fetus) OR (foetus)). To be included in the scoping review, articles needed to report studies using wearable devices to collect data about foetal and/or maternal characteristics during pregnancy. Peer-reviewed articles were included if they: (i) were written in English; (ii) involved human participants; (iii) used wearable devices; (iv) analysed foetal and/or maternal health parameters and characteristics; (v) were conducted during pregnancy. Published protocols were also included to investigate wearable devices, study designs, and expected findings of planned studies. Articles were excluded if they: (i) did not fit into the main question of the scoping review; (ii) only described the development of wearable devices without application on human participants; (iii) used sensors and/or mobile devices that were not wearable; (iv) did not analyse foetal or maternal parameters or activities; (v) were conducted before or after pregnancy. Abstracts, editorials, commentaries, reviews and systematic reviews, and meta-analyses were also excluded.

Two authors (AM and MB) independently selected the studies and charted the data, using a standardized data abstraction tool designed for this scoping review. The tool captured the relevant information on article characteristics (e.g., authors, publication year, country of origin), study characteristics (e.g., study design, characteristics of study population), detailed information on wearable device (e.g., type of device and sensors; parameters that were measured through the device), and the main findings achieved. Any disagreements were resolved through discussion between the two authors or further adjudication by a third author (AA).

Studies were grouped by the type of outcome considered (i.e., foetal or maternal parameters and characteristics), and summarized describing the type of setting, populations and study designs for each group, along with the wearable devices used and broad findings.

## 3. Results

### 3.1. Study Selection

After duplicates were removed, a total of 1244 articles were identified from literature databases. Based on screening of titles and the abstracts, 1119 were excluded while 125 full-text articles were retrieved and assessed for eligibility. Of these, 95 were excluded for the following reasons: 3 described the development or improvement of wearable devices without application on human participants; 5 were qualitative interview surveys on opinions and perceptions toward wearable devices; 53 used sensors that were not wearable; 1 did not analyse foetal and maternal parameters characteristics; 25 were conducted before or after pregnancy; 8 commentaries, reviews or systematic reviews. The remaining 30 studies were considered eligible for this scoping review (Figure 1) and their main characteristics (locations and study design) are summarized in Figure 2.

### 3.2. Wearable Sensors for Collecting Foetal Parameters

Nine studies described the application of wearable sensors to collect information about foetal parameters [12,13,14,15,16,17,18,19,20]. These studies are described in Table 1, together with a summary of their design and population, wearable device used, and main findings. The majority were from China (*n* = 4), followed by USA (*n* = 3), Australia and UK (*n* = 1, respectively). The vast majority of studies aimed at validating new wearable devices or systems for the collection of foetal parameters (*n* = 5), or at comparing different methods for data processing (*n* = 1). The remaining articles pertained to one case-control study and two prospective multicentre studies. Study populations ranged from 3 to 147 pregnant women at different weeks of gestation. The study by Yuan and colleagues, instead, used a maternal abdominal signal generator developed to simulate the abdominal surface signal of a pregnant woman [19]. In the following sections, we present the findings achieved by these studies, organized by the foetal parameter examined.

#### 3.2.1. Foetal Movements

Nearly half of studies detected foetal movements using accelerometer sensors (*n* = 3) or a combination of accelerometer and acoustic sensors (*n* = 1). The opportunity of detecting foetal movement through non-invasive sensor systems, in fact, has received a lot of attention for the research of foetal growth and development. However, recovering foetal movement signals from a continuous, low-amplitude, and heavily contaminated background is a challenging task, which requires advanced signal processing techniques to differentiate foetal movements from contaminating artefacts. To solve this issue, Liang and colleagues proposed a four-stage system that integrated foetal movement signal pre-processing, maternal artifact signal pre-identification, signal identification and classification [13]. The authors also compared the ability of two algorithms (i.e., orthogonal matching pursuit and adaptive filtering) to identify foetal movement signals, showing the best performance for the orthogonal matching pursuit algorithm [13]. In a second study, the authors refined their approach by combining the strength of Kalman filtering, time and frequency domain and wavelet domain feature extraction, and hyperparameter tuned Light Gradient Boosting Machine model [14]. In their comparative analysis of the Zenodo foetal movement dataset (i.e., a publicly available dataset of 16 different pregnant women), the novel approach exhibited better accuracy and robustness than existing methods for foetal movement signal recognition [14]. The same issue was faced by Mesbah and colleagues in their analysis of data from 21 pregnant women, collected through triaxial accelerometers [15]. The authors proposed an algorithm that combined independent component analysis for dimensionality reduction and discrete wavelet transform for artefact removal [15]. They also evaluated different classifiers showing the best performance for the Bagging classifier algorithm with random forest [15]. It is worth mentioning that the accuracy of systems described here ranged from 87.6% to 95.8%, indicating their great potential for prenatal foetal health monitoring. Further efforts to discriminate between different types of foetal movements have been made by Lai and colleagues, who proposed a wearable system based on a combination of accelerometers and bespoke acoustic sensors [12]. In their validation study, the authors evaluated a cohort of 44 pregnant women and demonstrated that the device was capable to discriminate vigorous foetal startle movements [12].

#### 3.2.2. Foetal Heart Rate

Three studies monitored foetal heart rate through devices that were based on different sets of electrodes (*n* = 2) or a device combining electrical and acoustic sensors (*n* = 1). In fact, the monitoring of changes in heart sounds, heart rate, and electrocardiogram (ECG) is useful to track foetal growth and for early detection of foetal congenital heart disease and distress. Yuan and colleagues proposed a foetal ECG monitoring device to detect maternal abdominal ECG signals, from which to extract foetal ECG signals and heart rate [19]. The authors tested the device using a maternal abdominal signal generator developed to simulate the abdominal surface signal of a pregnant woman. Their experimental results suggested that the device—which was also integrated with an app for Android smartphones—might be a feasible, non-invasive, monitoring system of foetal ECGs in real time [19]. Zhang and colleagues proposed a similar monitoring system with three electrodes, which was able to obtain foetal heart rate from the recorded abdominal ECG of pregnant women [20]. Their validation study was conducted on three pregnant women, showing good signal quality and high accuracy in three different postures (supine, seated, and standing) [20]. Mhajna and colleagues added to this knowledge, evaluating the performance of a wearable belt containing 8 electrical sensors and 4 acoustic sensors [17]. The comparison with data obtained with cardiotocography (i.e., the current standard in the healthcare setting) showed high correlations in a sample of 147 pregnant women [17]. In line with their findings, the authors proposed the device for a safe, non-invasive, and convenient monitoring of foetal and maternal heart rate in the clinic and remotely [17].

#### 3.2.3. Other Parameters

In a further study, Mhajna and colleagues used the wearable belt described above to monitor uterine activity before and during the delivery [16]. As part of pregnancy management, the uterine activity is currently monitored using an intrauterine pressure catheter or external tocodynamometer. However, the first one is an invasive approach that requires ruptured membranes, the second one is hampered by obesity, maternal movements, and belt positioning. Therefore, the authors conducted two separate prospective, comparative, open-label, multicentre studies to compare performances of the novel device against intrauterine pressure catheter and external tocodynamometer monitoring [16]. In both antepartum and intrapartum evaluations, the novel device provided more accurate and precise measurements than the current standards of care. Based on these findings, the novel device has a wide range of applications for monitoring both foetal heart rate and uterine activity [16]. The remaining study included in this section used a near-infrared spectroscopy device to evaluate placental oxygenation, a good indicator of placental structure and function. Nguyen and colleagues, in fact, developed a near-infrared spectroscopy device for the non-invasive transabdominal measurement of placental oxygenation [18]. The device was not just valid to measure placental oxygenation, but it was also useful to demonstrate lower placental oxygenation level in women with a complicated pregnancy [18].

### 3.3. Wearable Sensors for Collecting Maternal Parameters and Activities

Twenty-one studies described the application of wearable sensors to collect information about maternal parameters and activities during pregnancy [21,22,23,24,25,26,27,28,29,30,31,32,33,34,35,36,37,38,39,40,41]. These studies are described in Table 2, together with a summary of the design and population, wearable device used, and main findings. The majority were from USA (*n* = 9, including a study conducted in USA and Zambia) and Finland (*n* = 3). Other countries with only one study included: Australia, Bangladesh, Brazil, China, Israel, Japan, Perú, Singapore, and Taiwan. Six studies aimed at evaluating and/or validating wearable devices for the collection of maternal information: two case studies; two validation studies; and two feasibility studies. A total of seven observational prospective studies were conducted with the purpose of assessing the feasibility of using wearable devices during pregnancy and/or evaluating the relationships between maternal parameters, activities, and pregnancy outcomes. Three articles reported on randomized controlled trials, in which the efficacy of wearable-based interventions was evaluated. There were also two articles, one describing a semi-experimental study and the other describing a field trial. The remaining articles pertained to three protocols for planned prospective studies or randomized controlled trials, which evaluated the relationships between maternal activities and pregnancy outcomes. The majority of devices used in studies described above were wristband trackers of health parameters or activities (*n* = 16 studies), followed by finger-based health trackers (*n* = 2), body-conforming flexible wearable sensors (*n* = 2), and a system based on three chest, limb, and abdominal sensors (*n* = 1). Monitored information was varied and included maternal parameters (e.g., heart and respiratory rates, central and peripheral temperatures, blood oxygen saturation, blood pressure etc.) and activities (i.e., sleep characteristics, step count, and physical activity) during pregnancy. In the following sections, we present findings by grouping studies that used wearable sensors to exclusively monitor health parameters or maternal activities, and those that instead simultaneously collected both pieces of information.

#### 3.3.1. Cardiovascular Parameters

Five studies monitored a set of maternal parameters, in particular cardiovascular parameters and other clinical data, using wristband health trackers (*n* = 2), body-conforming flexible devices (*n* = 2), or a sensory system based on three sensors (*n* = 1). To our knowledge, the first example dated from 2005, when Maggioni and colleagues used an automated wearable device to monitor the circadian rhythm of maternal blood pressure during the third trimester. In particular, the authors found that a high variation in diastolic blood pressure was associated with intrauterine growth retardation [40]. Atzmon and colleagues conducted a prospective study evaluating changes in maternal parameters at delivery, through a wristband photoplethysmography monitoring device [21]. Continuous monitoring of maternal hemodynamic, in fact, could be crucial to ensure appropriate clinical care for all labouring women, and especially for those with heart disease, preeclampsia, or peripartum haemorrhage. Monitored parameters were cardiac output, blood pressure, stroke volume, systemic vascular resistance, and heart rate [21]. The authors reported that both epidural anaesthesia and delivery produced a slight increase in cardiac output [21]. With delivery, blood pressure increased slightly, reflecting the increase in cardiac output and the decrease in systemic vascular resistance. Placental expulsion was associated with a second peak in cardiac output and a decrease in systemic vascular resistance. These findings can be considered preliminary, as they were obtained from women without pregnancy complications [21]. Further studies should therefore monitor these hemodynamic parameters in labouring women with pre-existing cardiovascular and obstetrical complications. Ryu and colleagues addressed this issue, proposing a monitoring platform applicable across the entire continuum of antepartum and postpartum care [36]. The proposed system, based on three chest, limb, and abdominal sensors, provided non-invasive and continuous monitoring of heart and respiratory rates, pulse oxygenation, blood pressure, uterine electro-hysterography, and automated body position classification [36]. Study findings were evaluated in high-resource and low-resource settings, demonstrating the feasibility, performance, and acceptability of the system for both in-hospital and remote monitoring applications [36].

Another field of application of wearable devices is to capture physiological indicators of health, such as heart rate and its variability. In fact, wearable sensing could be a very important part of studies evaluating interventions to improve wellbeing and physiological stress during and after pregnancy. Ng and colleagues conducted a prospective study to predict physiological and perceived stress of pregnant women using a body-conforming flexible ECG sensor, ecological momentary assessment (EMA) surveys, and machine learning models [34]. Their results showed that it was possible to predict next-day physiological and perceived stress; however, sensor-based data alone had poor predictive performance and needed to be integrated with EMA-based data [34]. Another aspect to consider in these studies is the elevated burden for participants, which might affect the response to interventions. This issue was examined by Cummings and colleagues by evaluating the feasibility and acceptability of a body-conforming flexible ECG sensor and EMA to detect physiological stress and adjust the intervention accordingly [26]. Participants’ adherence was relatively low both for wearable device and EMA, especially among pregnant women with high perceived stress. Those with high household income, instead, were more likely to engage with the intervention content [26]. These considerations should therefore be kept in mind for the scalability and uptake of well-being interventions.

#### 3.3.2. Maternal Activities

Seven studies reported on the validity and feasibility of wearable devices used to monitor maternal activities during pregnancy, as well as on findings obtained through their application. All these studies were conducted using different types of wristband activity trackers. Kominiarek and colleagues conducted a study to assess the feasibility of a wristband activity tracker to measure objective physical activity among women in early pregnancy. Interestingly, objective measures of physical activity did not differ significantly from those collected through validated questionnaires. Moreover, pregnant women were greatly motivated to wear the device and reported high satisfaction with its use [41]. Ehrlich and colleagues examined the performance of a commercially available activity tracker in pregnant women with gestational diabetes mellitus (GDM) [27]. Their results suggested the device as a valid instrument to monitor walking or stepping-in-place, especially during the third trimester [27]. Similarly, Chen and colleagues compared data obtained through a commercially available activity tracker to self-reported physical activity levels [24]. The authors showed that the relationship between objective and self-reported data was non-linear, and that women were more engaged in physical activities in the second trimester. Objective physical activity levels, however, were not associated with the risk of developing GDM [24].

Wearable devices can also be useful to assess sleep quality during pregnancy. Accordingly, Galea and colleagues conducted a pilot study to evaluate the feasibility of objectively assessing sleep quality and the physical activity of pregnant women in a low-resource setting [28]. In spite of the promise of objectively monitoring sleep quality, the authors identified some challenges, including modest data completeness and participant acceptability [28].

As well as collecting data, wearable devices may be used to design interventions aimed at improving the behaviour of pregnant women. Kawajiri and colleagues employed a semi-experimental approach to evaluate the effects of an intervention against behaviour which was based on in-person advice, automatic alerts from wearable devices, and self-monitoring of sedentary behaviours [33]. In general, pregnant women accepted the intervention; however, it did not produce significant changes in their sedentary behaviour [33]. A similar approach was adopted by Cheung and colleagues in their randomized controlled trial evaluating the efficacy of an intervention, based on a wristband activity tracker integrated with text-messaging advice [25]. The study, conducted on pregnant women with GDM, demonstrated improvements in diet, physical activity, and weight gain. No significant changes were evident for glucose tolerance testing [25]. With a similar purpose, Chen and colleagues focused on the effects on gestational weight gain [23], which seriously affects other pregnancy and neonatal outcomes [42,43]. Their intervention consisted of using a wearable activity tracker and a mobile-health app among overweight and obese women [23]. The authors showed a significant lower proportion of women who exceeded their GWG in the intervention group, with the greatest effect observed during the second trimester [23].

#### 3.3.3. Simultaneous Collection of Cardiac Parameters and Maternal Activities

Clinical parameters and lifestyle data should be collected simultaneously in order to identify factors that contribute to adverse pregnancy outcomes and to hypothesize future interventions for pregnant women. Accordingly, six studies reported on the validity, feasibility, and perspectives of using wearable devices for the simultaneous monitoring of maternal health parameters and activities during pregnancy. In general, the devices described in this section were designed to monitor data on maternal parameters (e.g., heart rate and variability), physical activity (e.g., step count, used calories, stairs climbed, intensity of physical activity, etc.), and sleep patterns (e.g., total hours of sleep, sleep levels, and sleep movement). Controversies, however, existed principally around the feasibility of applying them in studies involving pregnant women. For instance, Nulty and colleagues explored the validity and acceptability of a commercially available activity tracker in pregnant and non-pregnant women [35]. Preliminary analyses conducted by the authors suggested that data collected through the activity tracker did not correlate well with criterion measures. Moreover, the acceptability of the device was higher in non-pregnant than in pregnant women [35]. By contrast, Sarhaddi and colleagues proposed a system, consisting of various data collectors and sensors, which was feasible and able to collect reliable photoplethysmography data and other information [38]. Grym and colleagues added to this evidence, evaluating the wearing time and experience with the use of a wristband activity tracker for women which was followed during the entire period of their pregnancy [29]. The authors reported that the daily use of the device was similar during the second and third trimesters, but then decreased after the delivery. The majority of participants, however, affirmed that the use of the device did not have long-lasting effects on behaviours [29]. Even though wearable devices did not appear to change behaviours, they may be useful to monitor maternal parameters and activities during pregnancy. Saarikko and colleagues, for example, showed how physical activity and sleep levels decreased from the second trimester to the third trimester [37]. By contrast, the average resting heart rate increased toward the third trimester and returned to the early pregnancy level during the postpartum period [37].

An alternative to wristband activity trackers has recently been tested on a few subjects, e.g., a finger-based activity tracker which was less invasive. This device is able to monitor resting heart rate and variability, sleep, and physical activity. In their first study, Jimah and colleagues reported significant correlations between objective and self-reported measures by analysing a pregnant woman at 33 weeks of gestation [31]. In a later study, these authors also reported physiological changes in resting heart rate and variability, respiratory, and sleep in two pregnant women after the onset of COVID-19 symptoms [32].

#### 3.3.4. Planned Studies

For the sake of completeness, the current scoping review included three protocols of already planned studies. The protocol proposed by Cai and colleagues regarded a prospective study which will use a wristband activity tracker to evaluate the association of physical activity level with gestational diabetes mellitus and hypertension [22]. Souza and colleagues have proposed the use of a wristband activity tracker to objectively monitor changes in physical activity and sleep patterns according to gestational age and pregnancy complications [39]. Finally, Hasan and colleagues have proposed a randomized controlled trial to investigate whether a wristband monitoring device could be helpful for monitoring blood pressure in pregnant women at risk of developing hypertensive disorders, and to achieve optimal maternal and foetal outcomes [30].

## 4. Discussion

In this scoping review, we have identified 27 primary studies and 3 protocols describing research using wearable devices to assess foetal and pregnancy outcomes. Except for one, these studies were published between 2018 and 2022, revealing how this field of research is relatively new and undergoing rapid and continual evolution. At its current stage, the research has focused primarily on the use of wearable devices for monitoring foetal vital signs (e.g., foetal heart rate and movements) and maternal activity during pregnancy (e.g., sleep patterns and physical activity levels).

In the first instance, included studies clearly demonstrated that non-invasive sensors, usually placed on the mother’s abdomen, are capable of detecting foetal heart rate and movements [12,13,14,15,16,17,19,20]. The task was not without objective difficulties, which arose when it became essential to differentiate between signals coming from the mother’s body and those coming from the foetus. The basis of every system consisted of accelerometers for foetal movements and electrodes for foetal heart rate. However, integration with acoustic sensors helped improve performance in either case [12,17]. Despite the high levels of performance and accuracy achieved, a number of other points were identified for improvement, including the use of different algorithms for data pre-processing, feature extraction, optimization, and classification [14]. This research, however, is still in its development and validation phases, and thus further work is required to overcome the main barriers to implementation.

For the second domain of application, selected studies reported on wearable devices used to collect maternal parameters and activities during pregnancy, and to investigate their association with maternal and neonatal health [21,22,23,24,25,26,27,28,29,30,31,32,33,34,35,36,37,38,39,40,41]. Devices used were varied, including wristband and finger-based trackers, body-conforming flexible sensors, and more complex systems made of different sensors. Included studies clearly showed the increasing interest in improving antenatal care with a non-invasive and continuous monitoring of maternal parameters in all trimesters of pregnancy and at delivery (e.g., heart and respiratory rates, pulse oxygenation, blood pressure, uterine electro-hysterography, etc.). Although promising in terms of improving the monitoring of maternal parameters, these devices have not yet produced many concrete results. Some studies reported high correlations between sensor-based measurements and those obtained using current monitoring technologies, which are too often expensive and complex [34,36]. However, these studies largely focused on uncomplicated pregnancies [21,26,34,36], while future research will have to be done on women with pregnancy complications. At the moment, the main fields of application appear to be those relating to the study of childbirth complications and physiological changes during pregnancy [21,26,34,36]. An additional aspect that should be improved is the compliance of participants with wearing wearable devices, which was quite low in some of the included studies [26].

This problem was also present in studies using wearable devices to monitor maternal activities during pregnancy, such as physical activity and sleep patterns. There was a general decreasing trend in participants’ compliance throughout the entire periconceptional period. In fact, compliance was higher before pregnancy [35], then decreased during the three trimesters, and dropped after delivery [29]. Despite this, some studies reported a good agreement between objective and subjective measures of maternal activities [24,27,28,41]. This was important, for example, to monitor changes in physical activity levels and sleep patterns across trimesters of pregnancy [24,28,37], and to investigate possible associations with pregnancy outcomes (e.g., GDM, stress, depression, etc.) [23,24,27,31,32]. In addition, it has been demonstrated how some interventions based on wearable devices and other innovative technologies may support healthy behaviours and adequate weight gain during pregnancy [23,25]. Some studies, however, did not demonstrate benefits from interventions, despite good acceptability and feasibility [29,33].

Overall, our findings clearly support and advocate the potential adoption of wearable devices for both antenatal care and research. However, our scoping review also indicates a paucity of research on specific pregnancy outcomes, as demonstrated by the limited number of studies on women with a complicated pregnancy. Likewise, only some interventions have proved effective in promoting adequate behaviours and the health of pregnant women. Effectiveness could be improved by increasing the compliance of participants with wearing the devices. Due to these considerations, improvements in the technical performance and structural design of wearable sensors, as well as adequate data processing, remain necessary [5,6]. This would be useful to enhance participants’ compliance and data quality. Measurements gained through sensors, in fact, should be robust as they can influence behaviours which can have unintended consequences on human health [6]. The design of such devices for health purposes must be undertaken with care, recognizing that users may choose not to follow recommendations concerning their use [6]. While some research groups already explored the possibility of acquiring data with a system of sensors [12,16,17,36], others used information obtained from other sources. The integration of sensor-based data with those collected through mobile health (mHealth) apps, for example, has proved to be suitable for improving the reliability of measurements [26,31]. The majority of mHealth apps reported in the selected studies were based on the so-called ecological momentary assessment approach [26,31], which allowed researchers to collect information about behaviours in real-time and in the real-world [44,45]. The use of data from social media, internet activities, Internet of Things (IoT) technologies, and specific surveys may also provide pertinent digital information to be considered [7,8,42,46,47,48,49,50,51]. Our scoping review also raised social and cultural issues, including concerns over data security and privacy, as well as fears about technology being intimately connected to the body. It was important to address these issues, especially as they relate to new-born health and safety. For this reason, several studies in the literature were conducted to understand clinicians’ and pregnant women’s perceptions of the potential benefits of mHealth in general and wearable technologies in particular [52,53,54,55]. Despite not currently using wearable devices in their medical practice, clinicians perceived the benefits of mHealth in supporting antenatal care [52,54]. It was also found that pregnant women were open to wearing wearables and using health monitoring devices, but were more likely to use them if clinicians were monitoring their data [52]. Furthermore, most women said they would change their behaviour if they received personalized recommendations on a smartphone during pregnancy [53]. In particular, clinicians and patients both expressed interest in monitoring foetal heart rate, blood pressure, and environmental exposure [54].

Some limitations should be considered when discussing our findings. The scope of our review covered a wide range of research conducted using wearable sensors during pregnancy. Moreover, we cannot exclude the existence of relevant studies, which were indexed in other databases (e.g., Scopus or Google Scholar). Accordingly, future systematic reviews should be encouraged to learn more about pitfalls and perspectives of using wearable devices in the antenatal care. Although there were many studies included that focused on wearable device development and/or validation, the evidence was generally obtained from studies involving a limited number of pregnant women, often without pregnancy complications. Moreover, there were few randomized controlled trials evaluating the effectiveness of wearable-based interventions.

## 5. Conclusions

Our scoping review identifies gaps in the literature and perspectives of wearable devices which may guide future research. Although promising and feasible, these technologies only partly address the needs of antenatal care in their current form. There is still insufficient evidence to describe the experience of pregnant women and to design effective interventions. Therefore, high quality research is needed to determine which and how wearable devices could support antenatal care.

## Figures and Tables

**Figure 1 jpm-13-00218-f001:**
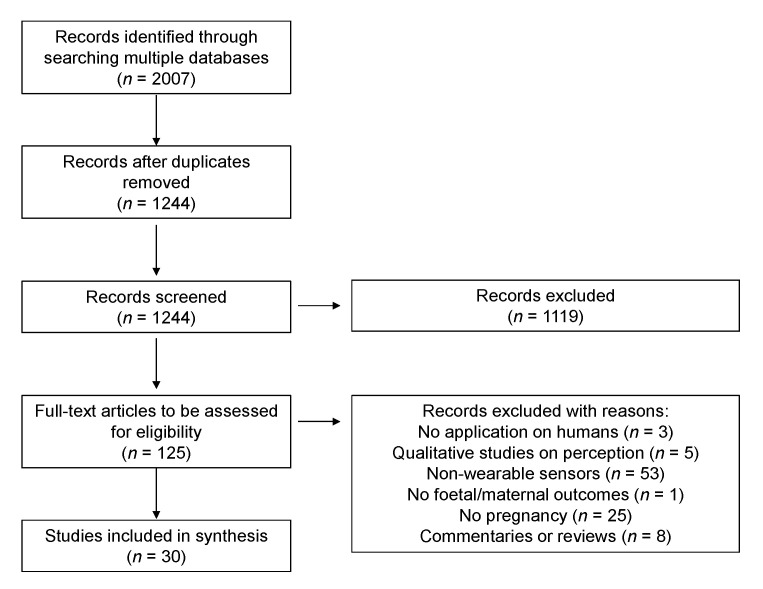
Selection of studies included in the scoping review.

**Figure 2 jpm-13-00218-f002:**
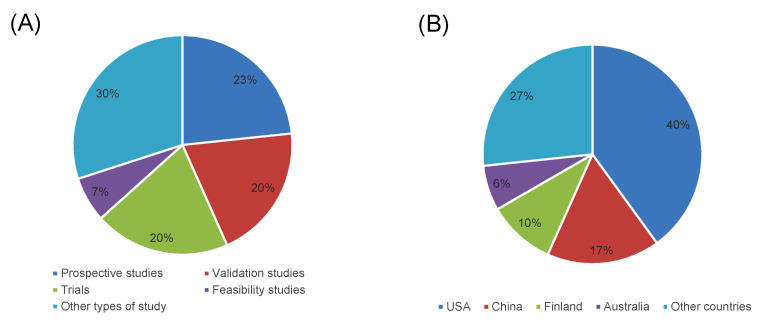
Percentages of studies included in the scoping review (*n* = 30) by location (**A**) and study design (**B**).

**Table 1 jpm-13-00218-t001:** Characteristics of studies using wearable devices for foetal characteristics and parameters.

First Author and Year of Publication	Country	Type of Study	Population	Wearable Design	Feature/s Captured	Main Findings
Lai et al., 2018 [12]	UK	Validation study	44 pregnant women between 25–36 weeks of gestation	Combination of accelerometers and bespoke acoustic sensors	Foetal movements	The device was able to discriminate startle movements from other forms of activity, and can effectively eliminate artefacts due to maternal movement
Liang et al., 2021 [13]	China	Validation study	4 pregnant women	Accelerometers	Foetal movements	The orthogonal matching pursuit algorithm was more effective than the adaptive filtering algorithm in identifying foetal movement signals
Liang et al., 2022 [14]	China	Comparative study	Publicly available dataset of 16 pregnant women	Accelerometers	Foetal movements	Compared with 8 existing methods for foetal movement signal recognition, the proposed method had better accuracy and robustness
Mesbah et al., 2021 [15]	Australia	Validation study	21 pregnant women with gestational age of at least 30 weeks	Tri-axial accelerometers	Foetal movements	The best performance was achieved by Bagging classifier algorithm, with random forest as its basis classifier
Mhajna et al., 2020 [16]	USA	Prospective, open-label, multicentre study	147 pregnant women with a mean gestational age of 37.7 weeks	Self-administered device consisting of 8 electrical sensors and 4 acoustic sensors	Foetal heart rate and maternal heart rate	Foetal heart rate measurements were highly correlated with cardiotocography. Maternal heart rate measured with the device was also highly correlated with that measured using cardiotocography
Mhajna et al., 2022 [17]	USA	Two separate prospective, comparative, open-label, multicentre studies	41 pregnant women with a mean gestational age of 38.8 weeks	Self-administered device consisting of 8 electrical sensors and 4 acoustic sensors	Uterine activity	In both groups (intrapartum and antepartum), the device had better sensitivity than tocodynamometry
Nguyen et al., 2021 [18]	USA	Case-control study	12 pregnant women with gestational age greater than 28 weeks (5 with pregnancy complications)	Near-Infrared Spectroscopy device	Placental oxygenation	Women with maternal pregnancy complications reported lower placental oxygenation level than those with uncomplicated pregnancy
Yuan et al., 2019 [19]	China	Validation study	Maternal abdominal signal generator developed to simulate the abdominal surface signal of a pregnant woman	Foetal electrocardiogram collector with five electrodes	Foetal heart rate	The proposed system may be feasible for non-invasive, real-time monitoring of foetal electrocardiogram
Zhang et al., 2022 [20]	China	Validation study	3 pregnant women	Foetal electrocardiogram monitoring system with three electrodes	Foetal heart rate	The proposed system had a promising application in foetal health monitoring

**Table 2 jpm-13-00218-t002:** Characteristics of studies using wearable devices for maternal parameters and activities.

First Author and Year of Publication	Country	Type of Study	Population	Wearable Design	Feature/s Captured
Atzmon et al., 2020 [21]	Israel	Prospective study	81 pregnant women at 37–42 gestational weeks	Wristband photoplethysmography monitoring device	Cardiac output, blood pressure, stroke volume, systemic vascular resistance, heart rate
Cai et al., 2019 [22]	Singapore	Protocol for a prospective study	408 women at <12 weeks of gestation	Wristband activity tracker	Step count
Chen et al., 2022 [23]	Taiwan	Randomized Controlled Trial	92 pregnant women assigned to the intervention and control groups	Wristband activity tracker	Step count
Chen et al., 2022b [24]	China	Prospective study	197 pregnant women at 10–14 gestational weeks	Wristband activity tracker	Objective physical activity
Cheung et al., 2019 [25]	Australia	Randomised Controlled Trial	60 pregnant women with gestational diabetes mellitus assigned to the intervention and control groups	Wristband activity tracker integrated with text-messaging	Objective physical activity
Cummings et al., 2022 [26]	USA	Randomized Controlled Trial	99 pregnant women at 18–22 gestational weeks assigned to the intervention and control group	Body-conforming flexible electrocardiograph sensors	Heart rate
Ehrlich et al., 2021 [27]	USA	Validation study	15 pregnant women with gestational diabetes mellitus and a mean gestational age of 32.8 weeks	Wristband activity tracker	Step count
Galea et al., 2020 [28]	Perú	Feasibility Study	13 pregnant women with a mean gestational age of 22 weeks	Wristband activity tracker	Step count and sleep characteristics
Grym et al., 2019 [29]	Finland	Prospective study	20 pregnant women at a median of 12.9 weeks of gestation	Wristband activity tracker	Step count, used calories, heart rate, stairs climbed, intensity of physical activity, total hours of sleep, sleep levels, sleep movements
Hasan et al., 2020 [30]	Bangladesh	Protocol for a Pilot Randomized Controlled Trial	70 pregnant women assigned to the intervention and control group	Wristband blood pressure monitoring device	Blood pressure
Jimah et al., 2021 [31]	USA	Case study	A pregnant woman at 33 weeks of gestation	Finger-based health tracker	Resting heart rate, resting heart rate variability, sleep, and physical activity
Jimah et al., 2022 [32]	USA	Case study	2 pregnant women with COVID-19	Finger-based health tracker	Resting heart rate, resting heart rate variability, sleep, and physical activity
Kawajiri et al., 2020 [33]	Japan	Semi-Experimental Study	56 pregnant women in the intervention group compared with an historical control group	Wristband activity tracker	Objective physical activity and sedentary behaviour
Kominiarek et al., 2019 [41]	USA	Feasibility Study	25 pregnant women at <16 weeks of gestation	Wristband activity tracker	Objective physical activity
Nulty et al., 2022 [35]	USA	Validation study	5 pregnant women	Wristband activity tracker	Step count, used calories, heart rate, stairs climbed, intensity of physical activity, total hours of sleep, sleep levels, sleep movements
Ryu et al., 2021 [36]	USA and Zambia	Field trial	576 pregnant women at 25–41 weeks of gestation	Maternal–foetal sensor system based on three chest, limb, and abdominal sensors	Maternal heart rate, respiratory rate, central temperature, SpO2, peripheral temperature, foetal heart rate, uterine contraction
Saarikko et al., 2020 [37]	Finland	Prospective study	20 pregnant women at ≤15 of weeks of gestation	Wristband activity tracker	Resting heart rate, resting heart rate variability, sleep, and physical activity
Sarhaddi et al., 2021 [38]	Finland	Prospective study	28 pregnant women at 12–15 gestational weeks	Wristband activity tracker	Resting heart rate, resting heart rate variability, sleep, and physical activity
Souza et al., 2019 [39]	Brazil	Protocol for a prospective study	400 pregnant women at 19–21 weeks of gestation	Wristband activity tracker	Objective physical activity and sleep pattern
Maggioni et al., 2005 [40]	USA	Prospective study	52 pregnant women during the third trimester	Automated wearable device	Blood pressure
Ng et al., 2022 [34]	USA	Prospective study	16 pregnant women at 10–18 weeks of gestation	Body-conforming flexible wearable sensor	Heart rate, heart rate variability

## Data Availability

Not applicable.

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
