# Peer review of "How Wearable Sensors Can Support the Research on Foetal and Pregnancy Outcomes: A Scoping Review"

_jpm, 2023, doi:10.3390/jpm13020218_

Round 1

Author Response

Dear Editor,

Please consider the revised version of the manuscript entitled “How wearable sensors can support the research on foetal and pregnancy outcomes: a scoping review” in which we have considered all comments and suggestions from reviewers. This letter is intended for the convenience of the editor and reviewers and contains the list of the requested changes. The following list of changes and answers to comments of Reviewers addresses all revisions made in the manuscript (in red font).

Reviewer 1

R: The authors conducted a scoping review to map the use of wearable devices or sensors during pregnancy to assess both foetal and maternal parameters. The manuscript is presented in a wellstructured manner and relevant for the field. The limitations of the articles included and the perspectives are well presented in the discussion section. Twenty eight studies were finally included in the scoping review. Not surprisingly, 19/28 concerned devices used to monitor maternal characteristics suggesting that research on monitoring fetal signs using wearable devices by the mother still requires development and validation phases.

Answer: We would like to take this opportunity to thank the Reviewer for his/her comments and suggestions which helped us in improving our manuscript.

R: My major comments would be on search strategy and terms use to identify the relevant articles: 1) The authors used only the combination of the terms (wearable) OR (wearables)) AND ((pregnancy) OR (pregnant)) to identify relevant articles in the different databases selected. The selection may seem very narrow in the first glance. Since, the terms “sensors” was used in the title of the article, what are the arguments not to include it in the search as well? Furthermore, other terms could have been used instead of “wearable”, like “mobile” or “personal”. Maybe other articles could have been selected in the scoping review using a broader search strategy? “Maternal” and “foetal” terms could have been added to the search strategy? 2) Articles were considered to be included in the scoping review if they were published between 2000 and October 2022. However, the 28 articles selected were all published between 2018 and 2022. The authors suggested that this very short period (4 years) is due “relatively new” development wearable devices (l.351, discussion section). Although, I completely agree with the authors that this field is in continual and rapid evolution, I was wondering how the limited number of terms used in the database search could have restricted the period of inclusion. If this field the in permanent evolution, so does the terms and vocabulary used. Would that be possible that back in 2000, other terms such as “mobile” or “personal” could have been used ?

A: As suggested, we have modified the search strategy including terms as mobile device, mother, fetus, etc. Please consider the revised version for all details. However, the use of terms such as mobile device and mother did not produce much more documents (we only included 2 additional articles), since the majority of retrieved studies was conducted using non-wearable devices and/or not during pregnancy. 

R: Minor comments: 1. Paragraph 3.3.2 (l.268-303) and paragraph 3.3.3 (l.304-336), both described “maternal activities” even though the last one also included maternal parameters. These two similar titles are confusing. Are the six studies described in the two paragraph different? Are the activities described in 3.3.2 that different from those described in 3.3.3 ? Are the parameters described in 3.3.3 may not be also described in the cardiovascular parameters in paragraph 3.3.1 ? âž” Please clarify the organisation of the 3.3 “wearable sensors for collecting maternal parameters and activities”.

A: Please consider the revised version, in which we have better clarified the organisation of the Results section.

R: L10-11 and l.49: I would have drop the term “systematically”

A: As suggested, we have removed the term systematically.

R: The objective of this scoping review is to “map about the feasibility and effectiveness of using wearable devices in the research of maternal and foetal outcomes”. However, all the studies included in the scoping did not assess the “feasibility and effectiveness” of such devices. Therefore, would that not be more correct to modify the objective as follows “map the use of wearable devices in the research of maternal and foetal outcomes” (l.11 & l.52).

A: As suggested, we have revised the aim of our study accordingly. 

Reviewer 2 Report

This work reviews how wearable sensors can support the research on both foetal and pregnancy outcomes. The work is timely. The English used is good. The paper has a good structure. I have the following minor comments for the authors to take into consideration to further improve the manuscript.

1. The organization of the paper/manuscript could be added at the end of Section 1. Eg. Section 2 presents… Section 3 discusses… Section 4 concludes…

2. Are there any other similar review works that have been conducted? If so, discuss the research gap. If none, highlight this as a novelty of this work.

3. Line 58/59: What are the justifications of using the database? Medical related? Reputable database? Scopus and Google scholar are not included in the list.

4. Some information (country, types of study) in Table 1 and 2 could be better visualized by presenting the info in suitable graphical representations, e.g. pie charts, bar charts.

5. Section 4 could be further structured into 4.1, 4.2, 4.3 with proper subheadings for better readability.

Author Response

Dear Editor,

Please consider the revised version of the manuscript entitled “How wearable sensors can support the research on foetal and pregnancy outcomes: a scoping review” in which we have considered all comments and suggestions from reviewers. This letter is intended for the convenience of the editor and reviewers and contains the list of the requested changes. The following list of changes and answers to comments of Reviewers addresses all revisions made in the manuscript (in red font).

Reviewer 2

R: This work reviews how wearable sensors can support the research on both foetal and pregnancy outcomes. The work is timely. The English used is good. The paper has a good structure. I have the following minor comments for the authors to take into consideration to further improve the manuscript.

Answer: We would like to take this opportunity to thank the Reviewer for his/her comments and suggestions which helped us in improving our manuscript.

R: Are there any other similar review works that have been conducted? If so, discuss the research gap. If none, highlight this as a novelty of this work.

A: We are grateful for this suggestion. There were several systematic about the role of mobile health in antenatal care. However, none of them focused on the use of wearable devices. We have added this point in the introduction section.

R: Line 58/59: What are the justifications of using the database? Medical related? Reputable database? Scopus and Google scholar are not included in the list.

A: In our scoping review, we used literature databases that are considered relevant in the medical field. As suggested, we have added this justification in the method section. Moreover, since we agree with the Reviewer that other databases could be consulted, we added this point as a limitation of our study.

R: Some information (country, types of study) in Table 1 and 2 could be better visualized by presenting the info in suitable graphical representations, e.g. pie charts, bar charts.

A: As suggested, we have added two pie charts (Figure 2) showing the percentage of studies included in the scoping review by location and study design.

R: Section 4 could be further structured into 4.1, 4.2, 4.3 with proper subheadings for better readability.

A: We agree with this comment, however, we followed the journal instructions, which stated that only the methods and results sections may be divided in subheadings.

Round 2

Reviewer 1 Report

Thank you very much for considering all the suggestions and modifications.